# Algae Oil Treatment Protects Retinal Ganglion Cells (RGCs) via ERK Signaling Pathway in Experimental Optic Nerve Ischemia

**DOI:** 10.3390/md18020083

**Published:** 2020-01-27

**Authors:** Tzu-Lun Huang, Yao-Tseng Wen, Yu-Chieh Ho, Jia-Kang Wang, Kuan-Hung Lin, Rong-Kung Tsai

**Affiliations:** 1Department of Ophthalmology, Far Eastern Memorial Hospital, Banqiao Dist., New Taipei City 220, Taiwanjiakangw@yahoo.com.tw (J.-K.W.); 2Department of Electrical Engineering, Yuan Ze University, Chung-Li, Taoyuan 320, Taiwan; 3Institute of Eye Research, Hualien Tzu Chi Hospital, Buddhist Tzu Chi Medical Foundation, Hualien 970, Taiwan; ytw193@gmail.com (Y.-T.W.); jasonho60912@gmail.com (Y.-C.H.); 4Department of Neurology, Taiwan Adventist Hospital, Taipei 105, Taiwan; kksao.lin@gmail.com; 5Institute of Medical Sciences, Tzu Chi University, Hualien 970, Taiwan

**Keywords:** rat model of anterior ischemic optic neuropathy, algae oil, nitric oxide synthase (iNOS), caspase 3, IL-1β, TNF-α, ciliary neurotrophic factor (CNTF), p-ERK

## Abstract

Background: We investigated the therapeutic effects and related mechanisms of algae oil (ALG) to protect retinal ganglion cells (RGCs) in a rat model of anterior ischemic optic neuropathy (rAION). Methods: Rats were daily gavaged with ALG after rAION induction for seven days. The therapeutic effects of ALG on rAION were evaluated using flash visual evoked potentials (FVEPs), retrograde labeling of RGCs, TUNEL assay of the retina, and ED1 staining of optic nerves (ONs). The levels of inducible nitric oxide synthase (iNOS), IL-1β, TNF-α, Cl-caspase-3, ciliary neurotrophic factor (CNTF), and p-ERK were analyzed by using western blots. Results: Protection of visual function in FVEPs amplitude was noted, with a better preservation of the P1–N2 amplitude in the ALG-treated group (*p* = 0.032) than in the rAION group. The density of RGCs was 2.4-fold higher in the ALG-treated group compared to that in the rAION group (*p* < 0.0001). The number of ED1-positive cells in ONs was significantly reduced 4.1-fold in the ALG-treated group compared to those in the rAION group (*p* = 0.029). The number of apoptotic RGCs was 3.2-fold lower in number in the ALG-treated group (*p* = 0.001) than that in the rAION group. The ALG treatment inhibited ERK activation to reduce the levels of iNOS, IL-1β, TNF-α, and Cl-caspase-3 and to increase the level of CNTF in the rAION model. Conclusion: The treatment with ALG after rAION induction inhibits ERK activation to provide both anti-inflammatory and antiapoptotic effects in rAION.

## 1. Introduction

The pathogenesis of nonarteritic anterior ischemic optic neuropathy (NAION) is caused by transient nonperfusion or hypoperfusion of an optic nerve head (ONH) [1]. Optic nerve (ON) ischemia induces a chain reaction of inflammation and edema, eventually resulting in retinal ganglion cell (RGC) death and vision loss [2]. Effective treatments for NAION are yet to be established. In a rat model of anterior ischemic optic neuropathy (rAION), breakdown of the blood optic-nerve barrier (BOB) was found to occur within hours’ postinduction of infarct [2], followed by early recruitment of extrinsic macrophages and activation of resident microglia at the core of an ischemic ON to trigger the proinflammatory cytokine-induced RGC apoptosis [3,4,5]. 

The potential of omega-3 polyunsaturated fatty acids (ω-3 PUFAs) including eicosapentaenoic acid (EPA, 20:5n-3) and docosahexaenoic acid (DHA, 22:6n-3) showed anti-inflammatory properties against ischemic injury, chronic inflammatory diseases, and cardiovascular risk, which has been investigated intensively [6,7,8,9,10]. Recently, there are few studies on the protective role of ω-3 PUFA supplementation with varying DHA/EPA ratios in both experimental and clinical investigations [11,12,13]. We previously reported that ω-3 PUFAs in fish oil had neuroprotective effects in rAION [4]. We found that the ratio of DHA to EPA (1:2) used in rAION-induced rats provided dual antiapoptotic and anti-inflammatory actions. Notably, one study demonstrated that pure DHA or a combination containing more DHA was stronger than EPA to promote the expression of neurotrophins and their receptors in neuron cell lines [11]. Moreover, both DHA and EPA attenuated apoptosis and improved cell viability, but DHA was more effective than EPA [11]. Supplementation with high-DHA fish oil, in comparison with placebo and high-EPA fish oil supplements, significantly increased concentrations of oxygenated haemoglobin and total levels of haemoglobin, indicating an increase in the cerebral blood flow [14]. It was also reported that DHA had a much higher tendency to accumulate into sphingomyelin/cholesterol-rich raft-like domains on membranes than EPA, making it having a much higher potential to affect cell signaling by modifying the structures and compositions of these lipid rafts [15]. Taken together, we considered that higher content of DHA might provide better protective effects for ischemic optic neuropathy. Traditional fish oils contained higher amounts of EPA than DHA [16]. Bioengineered marine microalgae *Schizochytrium* sp. is currently used to produce DHA-rich ω-3 PUFAs supplements [17,18]. Alternative ω-3 PUFAs supplements to fish oil seems to be a crucial nutritional issue because of the vegetarian’s concern and low DHA content [19]. Therefore, we chose marine algae oil to investigate the neuroprotection effects of DHA-rich ω-3 PUFAs in a model of rAION.

Evidences have demonstrated that ω-3 PUFAs can modulate different signaling pathways to inhibit cell apoptosis and inflammation in many experimental models [16,20]. For neuroinflammation and neuroprotection, COX2 and NF-κB are the most widely studied signaling pathways in many in vivo and in vitro studies of ω-3 PUFAs treatment [20]. Recent studies reported that ω-3 PUFAs can regulate the ERK signaling pathway in different cancer cells [21,22,23]. However, it is little known for the role of the ERK signaling pathway in ON injuries. Previous reports have demonstrated that the inhibition of the ERK pathway reduced the levels of proinflammatory cytokines and led to neuroprotection in brain injuries [23,24,25,26]. Thus, we considered that ω-3 PUFAs may regulate the ERK signaling pathway to provide neuroprotective effects in an rAION model. 

In the present study, we evaluated the protective effects of marine algae oil on visual function, RGC apoptosis, and neuroinflammation in an rAION model. Furthermore, we evaluated the role of the ERK signaling pathway in the antiapoptotic effect and anti-inflammatory effects. Thus, it is the first study to evaluate the protective role of the ω-3 PUFAs-regulated ERK signaling pathway in ON ischemia.

## 2. Results

### 2.1. Determination of the Blood AA/EPA Ratio

During the daily gavage with algae oil, the blood AA/EPA (a marker of cellular inflammation by following two essential fatty acids) ratio gradually decreased from 17.30 ± 2.1 to 0.76 ± 0.25 (Figure 1). The AA/EPA ratio stably dropped below 1.0, following daily gavage with algae oil for six days. Therefore, we decided to use daily gavage with algae oil for seven days as a treatment duration in this study.

### 2.2. Treatment with Algae Oil Preserves Visual Function

Changes in flash visual-evoked potentials (FVEPs; visual function assessment) after rAION were measured in the 4th week after infarct. The amplitude of the P1–N2 wavelet was used to evaluate the RGC structure and function in vivo. The RGC loss resulted in the P1–N2 amplitude decrease. A previous study showed no significant differences in latency between photoptic and scotopic FVEPs in Wistar rats [5]. In this study, we determined the amplitude of the P1–N2 wavelet in a sham group, which was 64 ± 11 μV. The amplitudes of the P1–N2 waves in the phosphate-buffered saline (PBS)-treated group and the algae-oil-treated group were 20 ± 6 and 41 ± 13 μV, respectively (Figure 2). There was a significant preservation of amplitude in the algae-oil-treated group as compared with that of the PBS-treated group (*p* = 0.032).

### 2.3. Treatment with Algae Oil Preserves RGC Density

The RGC density in the central retinas of the sham-operated group was 1479 ± 46 /mm^2^. At the 4th week postinfarct, the central RGC density in the PBS-treated group decreased to 506 ± 226 /mm^2^. In the algae-oil-treated group, the RGC density also declined by a smaller amount to 1215 ± 504 /mm^2^ (*p* < 0.0001 when compared to that in the PBS-treated group, *n* = 12 per group) (Figure 3).

### 2.4. Treatment with Algae Oil Inhibites Extrinsic Vascular-Borne Macrophage Infiltration and the Expression Levels of Inducible Nitric Oxide Synthase (iNOSs), TNF-α, IL-1β, and p-ERK ½

In ED1 (marker of CD38, specific for extrinsic macrophages) staining of ON sections, there were 3.1 ± 1.0 ED1^+^ cells per high-power field (HPF) in the sham group. After rAION induction, the number of ED1^+^ cells infiltrating the injured area of the ON was increased in the PBS-treated group (26.6 ± 9.4 cells per HPF). The number of ED1^+^ cell infiltration in the ON was shown as 5.5 ± 3.5 cells per HPF in the algae-oil-treated group (Figure 4A). The number of ED1^+^ cell was significantly lower in the algae-oil-treated group than that in the PBS-treated group (Figure 4B; *p* = 0.029). In addition, significantly decreased levels of proinflammatory cytokines were found in the algae-oil-treated group compared to in the PBS-treated group (Figure 4C). The levels of iNOS, TNF-α, and IL-1β were decreased 3.95-, 2.11-, and 3.29-fold, respectively, in the algae-oil-treated group compared to those in the PBS-treated group (Figure 4D, *p* < 0.05). Notably, p-ERK2 was highly decreased 7.15-fold in the algae-oil-treated group compared to that in the PBS-treated group (Figure 4D; *p* < 0.05).

### 2.5. Treatment with Algae Oil Reduces Apoptotic RGCs and the Levels of Cl-Caspase-3 but Increases the Level of Ciliary Neurotrophic Factor (CNTF)

There were very few TUNEL-positive cells in the sham group (Figure 5A). The TUNEL-positive cells in the RGC layer were significantly decreased in the algae-oil-treated group to 2.3 ± 2.7 cells per HPF (Figure 5B; *p* = 0.001, compared to in the PBS-treated group (7.4 ± 2.4 cells per HPF)). 

Additionally, the level of Cl-caspase-3 was 9.59-fold lower in the algae-oil-treated group than that in the PBS-treated group (Figure 5C,D; *p* < 0.05). Moreover, the level of CNTF was 2.35-fold higher in the algae-oil-treated group compared to that in the PBS-treated group (Figure 5C,D; *p* < 0.05).

### 2.6. Treatment with an ERK Inhibitor Reduces the Levels of iNOS, TNF-α, IL-1β, and Cl-Caspase-3 but Increases the Level of CNTF

The levels of iNOS, TNF-α, IL-1β, and Cl-caspase-3 were decreased 6.85-, 4-, 2.05-, and 9.5-fold, respectively, in the ERK inhibitor (PD98059)-treated group compared to those in the PBS-treated group (Figure 6; *p* < 0.05). In addition, the level of CNTF was 4.93-fold higher in the ERK-inhibitor-treated group than that in the PBS-treated group (Figure 6; *p* < 0.05).

## 3. Discussion

The present study proved that the seven-day continuous algae oil treatment significantly suppressed the macrophage infiltration at ONs and lessened RGCs death, as well as providing better visual function preservation in rAION. We further confirmed that algae oil successfully ameliorated ERK expression to inhibit the expression levels of injury-induced proinflammatory cytokines (iNOS, IL-1β, and TNF-α) and an apoptotic factor (Cl-caspase-3) and increased the expression level of a neurotrophic factor (CNTF).

AA/EPA ratios reaching 11.1–2.1 were associated with a less level of proinflammatory productions [27]. A lower AA/EPA ratio in serum was essential for reducing proinflammatory cytokines, and a higher AA/EPA ratio was noted in age-related macular degeneration [28,29]. Thus, we investigated that continue dosage of algae oil resulted in an AA/EPA ratio less than 1.5 on the 6th day of gavage, which represented the optimal AA/EPA ratio in serum for anti-inflammation in rAION. That is the rationality for our therapeutic protocol of seven-day algae oil gavage. In our previous report, the AA/EPA ratio stably dropped to 1.5, following daily gavage with fish oil (Omega Rx Zone) for 10 days, which may explain shorter treatment duration with algae oil can provide the same anti-inflammatory and protective effects as the fish oil treatment in rAION [4]. Fish oil consisted of purified ethyl esters rich in EPA (400 mg/g) and DHA (200 mg/g); however, our algae oil consisted of concentrated triglyceride rich in EPA (317 mg/g) and DHA (556 mg/g), and the concentration of DHA in algae oil was higher than in fish oil. A previous study has shown that triglyceride-form ω-3 PUFAs was better absorbed than synthetic ethyl ester-form ω-3 PUFAs [30]. They also found that the AA/EPA ratio was lower after concentrated TG supplementation than ethyl ester supplementation [30]. Similar findings were observed in our studies that the treatment with algae oil spent shorter duration in reducing the AA/EPA ratio to 1.5 than the treatment with fish oil. We considered that this shorter treatment duration with algae oil was because of the different bioavailability of two forms of ω-3 PUFAs. It also reflected that triglyceride-form ω-3 PUFAs might provide a better anti-inflammatory effect for acute inflammation than ethyl-ester-form ω-3 PUFAs. 

Noteworthy, the concentration of DHA in algae oil is higher than that of fish oil. Why does higher DHA concentration cause stronger neuroprotective effects in the occurrence of ischemia? The promising effect results from the ability of two metabolic products of DHA, namely maresin 1 and resolvin D1, which promote the switch of macrophage M1 to the M2 phenotype, and the activated M2 phenotype of microglia/macrophages have neuroprotective effects [31,32]. The existence of resolvin D2 was discovered in the improvement of ischemic lesions in a model of brain stroke [33]. Recent research has elucidated that resolvins are derived from DHA, which have shown better anti-inflammatory activity than other EPA-derived eicosanoids in immune cells [34]. From the above evidence, we considered that algae oil had better effects of anti-inflammation and neuroprotection than fish oil.

In this study, the levels of ERK and proinflammatory cytokines were decreased in the algae-oil-treated group compared to those in the rAION-inducted group. In addition, the ERK inhibition reduced the expression levels of iNOS, IL-1β, and TNF-α in the rAION model. These data demonstrated that algae oil modulated the expression levels of pro-inflammatory cytokines through ERK inhibition. A previous study reported similar evidence that DHA treatment in macrophage culture can modulate the levels of proinflammatory mediators by inhibiting ERK activation [35]. Some in vitro studies reported that the ERK1/2 activation resulted in the inflammatory response in microglia [36,37,38]. Therefore, suppression of the ERK signaling pathway becomes a pharmaceutical-based strategy to reduce neuroinflammation in stroke, neurodegenerative disorders, intracranial infections, and other diseases [39,40,41]. The ERK signaling pathway is also known to mediate the regulation of inflammatory responses, cytokines, and cell apoptosis and death in ischemic and hemorrhagic brain injury [42,43,44,45,46]. Taken together, we considered that the DHA-rich ω-3 PUFAs can reduce the levels of proinflammatory cytokines after ON infarct via the ERK-dependent signaling pathway. 

Our observations confirmed that the algae oil treatment successfully decreased the apoptosis of RGCs based on the data of the TUNEL assay and the decrease level of Cl-caspase-3 in the retinal samples. Additionally, the level of ERK was significantly reduced by treatment with algae oil after ON infarct. We also found the treatment with an ERK inhibitor decreased that level of Cl-caspase-3 in the rAION model. Caspases that are proapoptotic factors can lead to cell death, which can mediate cytokines such as IL-1β and TNF-α [47]. In this study, the treatment with algae oil can reduce the levels of proinflammatory cytokines, which may inhibit cytokine-induced cell death in the rAION model. Moreover, a previous study also demonstrated that ERK activation promoted hydrogen-peroxide-induced apoptosis through Cl-caspase-3 activation and inhibition of Akt in renal epithelial cells [48]. Another study reported that ERK inhibitors attenuated ischemia/reperfusion-induced apoptosis in the myocardium [49]. Herein, we demonstrated that DHA-rich ω-3 PUFAs may reduce ERK expression to inhibit capapse-3-dependent apoptosis. 

The level of CNTF in the retinas was higher in the algae-oil-treated group compared with that in the PBS-treated group after ischemic insult. DHA is a precursor for the synthesis of neuroprotectin D1, promoting neuronal and/or photoreceptor cell survival factors associated with several neurotrophic factors, including CNTF [50,51]. In addition, we found that ERK inhibition can induce CNTF expression after ON infarct. A previous report demonstrated that an inactive state of ERK was crucial for the CNTF expression in Schwann cells and the activation of ERK following nerve injury critically influenced the expression of CNTF [52]. CNTF also activated PI3K/AKT signaling [52,53] and has been identified as an essential mediator for neurite outgrowth of mature RGCs and neuroprotection of inflammatory stimulation in the eye [54]. In addition, the role of CNTF on temporary survival improvement of RGCs has been proven, when intravitreal injection of CNTF was applied within one day in a model of rAION [55]. Taken together, we considered that DHA-rich ω-3 PUFAs can inhibit ERK activation to increase the level of CNTF in an rAION model. 

In conclusion, we demonstrated that continuous daily oral gavage of algae oil for seven days after rAION could protect RGCs from death, indicating that algae oil has neuroprotective effects via multiple actions, including antiapoptosis of RGCs and anti-inflammation, as well as inducing expression of a neurotrophic factor in the retina (Figure 7). Additionally, DHA-rich algae oil can modulate the expression levels of proinflammatory cytokines, Cl-caspase-3, and CTNF via the ERK signaling pathway in the rAION model (Figure 7). Even though we proved that the DHA-rich algae oil provided neuroprotective effects in rAION, the other ingredients in commercial algae oil may also provide some biological effects in this study. This shortcoming of this study may result in overestimation/underestimation of the neuroprotective effects of algae oil in rAION. Further investigation is needed to reveal the functions of these ingredients in an rAION model.

## 4. Materials and Methods 

### 4.1. Animals

Adult male Wistar rats weighing 150–180 g (7–8 weeks old) were used in this study. The rats were obtained from the breeding colony of BioLASCO Co., Taipei, Taiwan. The animal care and experimental procedures were performed in accordance with the Statement for the Use of Animals in Ophthalmic and Vision Research. In addition, the Institutional Animal Care and Use Committee of Buddhist Tzu Chi General Hospital approved all animal experiments. 

### 4.2. Study Design

In the present study, we investigated the therapeutic effect of marine algae oil (from *Schizochytrium* sp., Nordic Natural, NC, Watsonvilla, CA, USA) in an rAION model; the marine algae oil is certified by the American Vegetarian Association. The content of algae oil included eicosapentaenoic acid (EPA) 200 mg/g and docosahexaenoic acid (DHA) 350 mg/g for the liquid formulation. To monitor the blood AA/EPA ratio, six wild-type rats were administered marine algae oil (EPA: 1 g/day; DHA: 1.75 g/day) daily for 10 days. During the daily gavage, the blood samples were collected once daily to determine the AA/EPA ratio. We found that the AA/EPA ratio was less than 1.5 after 6 days of gavage. 

In the following rAION experiments, 36 rats received rAION treatment and received a gavage administration once daily of either algae oil (1 μL/Kg body weight per day, *n* = 12) (rAION + ALG group) or PBS (serving as the control; *n* = 12) starting immediately after rAION induction for 7 consecutive days. Another sham group received laser treatment without the use of photosensitizing agents (*n* = 12). All the animals tolerated this treatment without any complications, and all of them survived until the end of the treatment. The rats were euthanized using CO_2_ insufflation 4 weeks’ postinfarct. 

For investigating the signaling pathway, 6 rAION-inducted rats were intravitreally injected with 5 μL of an ERK inhibitor (PD98059, Sigma-Aldrich, St. Louis, MO, USA) and another 6 rAION-inducted rats were intravitreally injected with 5 μL of PBS. The retina samples were collected for immunoblotting analysis at day 3 post-rAION.

### 4.3. rAION Induction 

The rAION induction protocol followed our previous report [56]. Briefly, after general anesthesia, rose bengal (RB) (Sigma-Aldrich, St. Louis, MO, USA) was administered intravenously through the tail vein (2.5 mM RB in PBS and 1 mL per animal weight). The sham laser treatment consisted of illuminating the ONH region with an argon laser without RB injection. After administration of RB and pupil dilation, the right optical disc of the rat was directly treated with an argon green laser at wavelengths of 532 nm and 500 μm and a power of 80 mW (MC-500 multicolor laser, Nidek Co., Ltd, Tokyo, Japan) with 12 pulses and 1 s duration each by a fundus contact lens. The laser settings were the same as those in our previous paper [5].

### 4.4. FVEPs

The detailed procedures of recording FVEPs were described in our previous reports [56,57,58]. Briefly, we used a visual electrodiagnostic system (Diagnosys LLC, Lowell, USA) to measure FVEPs. To exclude the possibility of the contralateral eye contaminating the FVEP test, we covered the fellow eye while performing the stimulation. We compared the amplitude of the P1–N2 wave in each group to evaluate visual function (*n* = 12 rats per group).

### 4.5. Retrograde Labeling of RGCs with FluoroGold and Morphometry of the RGCs

The detailed protocol of FluoroGold labeling was described in our previous reports [58,59]. In brief, retrograde labeling was performed 1 week before rats were euthanized. Rats were anesthetized and placed on a stereotactic frame; 2 μL 5% of FluoroGold (Fluorochrome LLC, Denver, CO, USA) was injected into the superior colliculus (AP: −6mm; ML: −1.5mm; DV: 4mm) on each side. After sacrifice of rats, eyeballs were collected and fixed in 10% formalin for 1 h, and then flat-mounted retinas were examined with a 200× fluorescence microscope (Axioplan 2 imaging, Carl Zeiss, NewYork, USA). RGC density was counted using GE ImageMaster 2D software (GE Healthcare Biosciences, Uppsala, Sweden). The retinas were examined for RGCs at a distance of 1 mm from the center to provide central RGC densities. We counted at least eight randomly chosen areas in the central regions (about 40% of the central area) of each retina, and their averages were used as the mean density of central RGCs per retina (*n* = 6 rats per group). 

### 4.6. In Situ Nick End-Labeling (TUNEL) Assay

To ensure the use of equivalent fields for comparison, all the frozen retinal sections were prepared with a 1–2 mm distance from the ON head. We counted apoptotic cells using the TUNEL assay kit (Click-iT™ Plus TUNEL Assay, Invitrogen, Waltham, USA ). Nuclei were stained with 4t,6-diamidino-2-phenylindole (DAPI). The TUNEL-positive cells in the RGC layer of each sample were counted in 10 HPF (400×), [57,58,59], and an average of three sections per retina was used for further analysis (*n* = 6 rats per group). 

### 4.7. Immunohistochemistry (IHC) of ED1-Positive Cell in the ON

The detailed procedure for ED1 labeling was described in our previous reports [57,58,59]. ONs were embedded in an OCT compound (Tissue-Tek, Sakura, Torrance, USA), and 20 µm thick longitudinal sectioning was performed using a cryostat (Leica Microsystems, Wetzlar, Germany). The cryosections were blocked in PBST with 5% BSA at room temperature for 1 hour. A primary antibody ED1 (1:50; Bio-RAD, Hercules, USA) was applied in 4 °C overnight. Secondary antibodies were then incubated and conjugated with Alexa Fluor 555 (1:200; Life Technology, Carlsbad, USA) at room temperature for 1 hour. Sections were then counterstained with DAPI (1:200; Sigma-Aldrich, Louis, USA). ED1-positive cells were quantified using ImageJ (https://imagej.nih.gov/ij/) by measuring the fluorescence intensity. For comparison, ED1-positive cells were counted in six HPFs (magnification: 400×) at ON lesion sites (*n* = 6 rats per group).

### 4.8. Western Blotting

Retinas were lysed in a RIPA buffer (EMD Merck, Kenilworth, USA) containing a phosphatase inhibitor PhosSTOP (Roche, Branchburg, USA). Protein was separated on NuPAGE gels (Life Technologies, Carlsbad, USA) and transferred to polyvinylidene difluoride membranes (Life Technologies). Blots were blocked with 5% nonfat milk in a TBST buffer (0.02 M Tris-base (pH 7.6), 0.8% NaCl, 0.1% Tween 20) for 60 min. After rinsing with the TBST buffer, samples were incubated with primary antibodies at 4 °C overnight. The antibodies we used were as follows: iNOS (1:500, Abcam, Cambridge, UK), Cl-caspase-3 (1:1000, Cell Signaling, Danvers, USA), IL-1β (1:250, Novus Biologicals, Littleton, USA), TNF-α (1:500, Abcam), CNTF (1:3000, Abcam), p-ERK (1:1000, Cell Signaling), and GAPDH (1:5000, Cell Signaling). Membranes were washed twice with the TBST buffer, followed by incubation with appropriate Horseradish Peroxidase (HRP)-conjugated secondary antibodies (1:10000, Jackson Immunoresearch, PA, USA) at room temperature for 1 hour and detected with Immobilon® ECL Ultra Western HRP Substrate (Millipore, Billerica, USA). Results were quantified using ImageJ software (*n* = 6 in each group).

### 4.9. Statistical Analysis

All the measurements were performed in a blinded fashion, including histological interpretation. Statistical analysis was performed using a commercial software package (Graphpad Prism) (IBM SPSS Statistics 19, International Business Machines Corp., Armonk, NY, USA). We used the Mann–Whitney U test to evaluate differences in the number of cells between groups. Results with *p* values less than 0.05 were considered statistically significant. 

## Figures and Tables

**Figure 1 marinedrugs-18-00083-f001:**
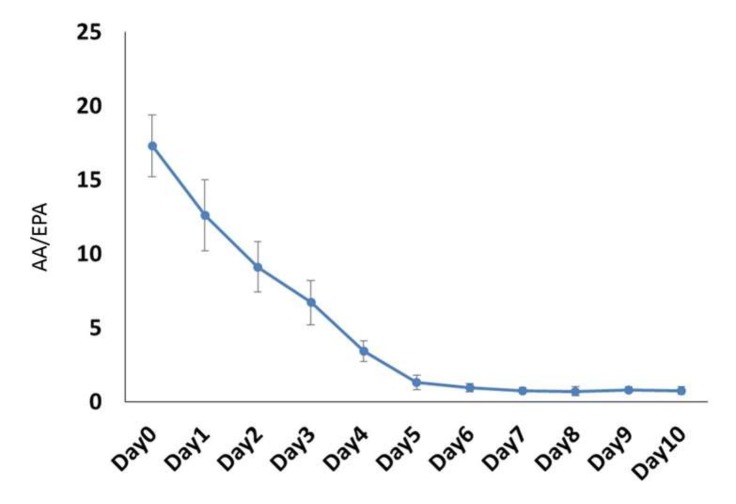
The change of the level of blood AA/EPA ratio from day 0 to 10 after feeding with algae oil (AGL). Six adult male Wistar rats were gavaged with algae oil once daily and monitored for their daily blood AA/EPA ratio.

**Figure 2 marinedrugs-18-00083-f002:**
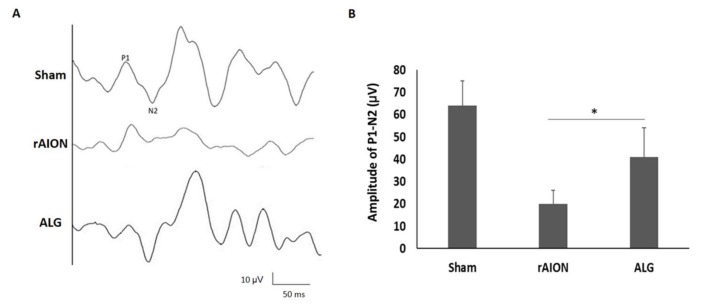
Evaluation of visual functional assessments through flash visual-evoked potential (FVEP) recordings in the 4th week after infarct: (**A**) representative FVEP wavelet in each group in a rat model of anterior ischemic optic neuropathy (rAION); (**B**) bar chart showing the P1–N2 amplitudes. There was a significant improvement in the FVEP in the ALG-treated group as compared to that in the phosphate-buffered saline (PBS)-treated group (rAION) (* *p* = 0.032, *n* =12). Data are expressed as mean ± SD.

**Figure 3 marinedrugs-18-00083-f003:**
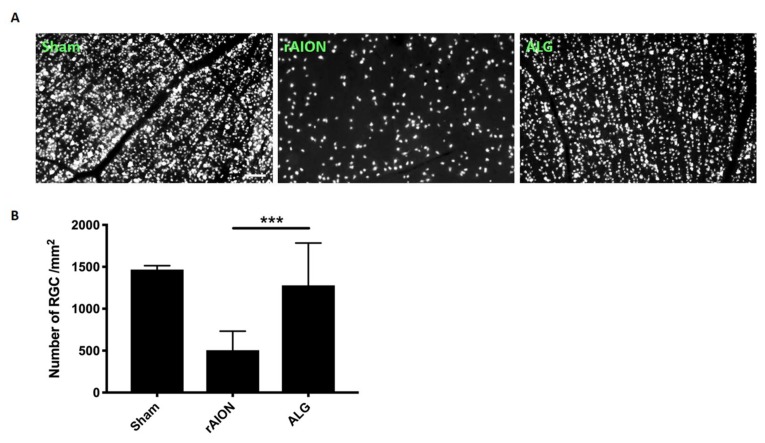
Morphometry of retinal ganglion cells (RGCs) in rAION-induced rats with PBS or ALG treatment: (**A**) representative RGC density in the central retinas in each group; (**B**) bar chart showing the RGC density in the AGL-treated group declined less in the central retina compared to in the PBS-treated group (*** *p* < 0.0001, *n* = 12 per group; scale bar = 50 μm).

**Figure 4 marinedrugs-18-00083-f004:**
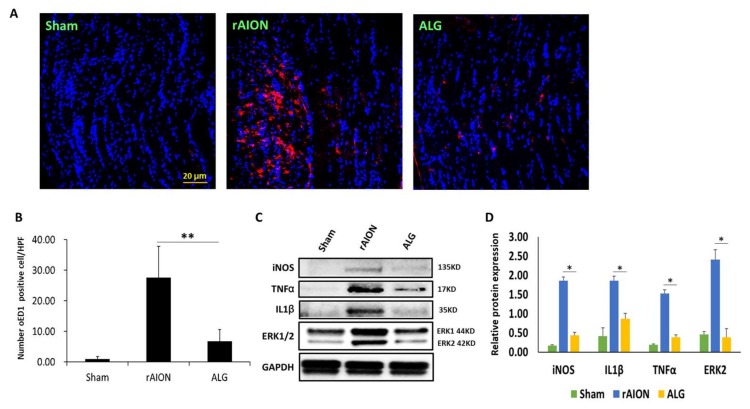
Immunohistochemistry of ED1 in optic nerves (ONs) four weeks after rAION induction to evaluate blood-borne macrophage infiltration: (**A**) representative image of ED1 staining in longitudinal sections of the ONs. The ED1-positive cells and the nuclei are marked in red and blue colors, respectively; (**B**) quantification of ED1^+^ cells per high-power field (HPF). Data are expressed as mean ± SD (*n* = 6 per group). The decreasing number of ED1^+^ cells was significantly different in the algae-oil-treated group as compared to in the PBS-treated group (** *p* = 0.029; *n* = 6; scale bar = 20 μm); (**C**) western blot analysis of inducible nitric oxide synthase (iNOS), IL-1β, TNF-α and ERK expression of the rat retina after anterior ischemic optic neuropathy (AION) on the 3rd day; (**D**) bar chart showing the levels of iNOS, TNF-α, and IL-1β were decreased 3.95-, 2.11-, and 3.29-fold, respectively in the AGL-treated group compared to those in the PBS-treated group (* *p* < 0.05). p-ERK2 was highly decreased 7.15-fold in the AGL-treated group compared to that in the PBS-treated group (* *p* < 0.05; *n* = 6 per group).

**Figure 5 marinedrugs-18-00083-f005:**
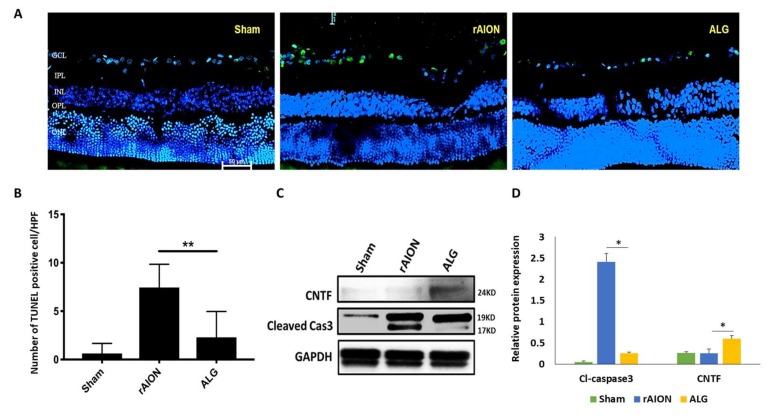
Analysis of RGC apoptosis in anthe RGC layer through the TUNEL assay in the 4th week after rAION induction: (**A**) representative image of TUNEL staining; (**B**) quantification of apoptotic cells per HPF. The TUNEL-positive cells in the RGC layer significantly decreased in number in the rAION + ALG group (** *p* < 0.005, *n* = 6; scale bar = 50 μm); (**C**) western blot analysis of Cl-caspase-3 and ciliary neurotrophic factor (CNTF) expression of the rat retina after AION on the 3rd day; (**D**) bar chart showing the level of Cl-caspase-3 was decreased 9.59-fold in the AGL-treated group compared to in the PBS-treated group (* *p* < 0.05). The treatment with AGL induced a 2.35-fold increase in the level of CNTF compared to that in the PBS treatment (* *p* = 0.012, *n* = 6). Abbreviations: GCL, ganglion cell layer; IPL, inner plexiform layer; INL, inner nuclear layer; OPL, outer plexiform layer; ONL, outer nuclear layer.

**Figure 6 marinedrugs-18-00083-f006:**
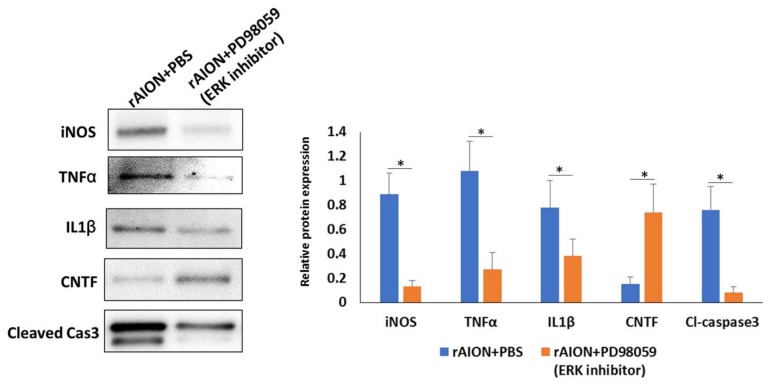
Western blot analysis of iNOS, TNF-α, IL-1β, Cl-caspase-3, and CNTF expression in the group of rAION treated with PBS and an ERK inhibitor (PD98059). The levels of iNOS, TNF-α, IL-1β, and Cl-caspase-3 were decreased 6.85-, 4-, 2.05-, and 9.5-fold, respectively, in the ERK inhibitor (PD98059)-treated group compared to those in the PBS-treated group (* *p* < 0.05). In addition, the level of CNTF was 4.93-fold higher in the ERK-inhibitor-treated group than that in the PBS-treated group (* *p* < 0.05).

**Figure 7 marinedrugs-18-00083-f007:**
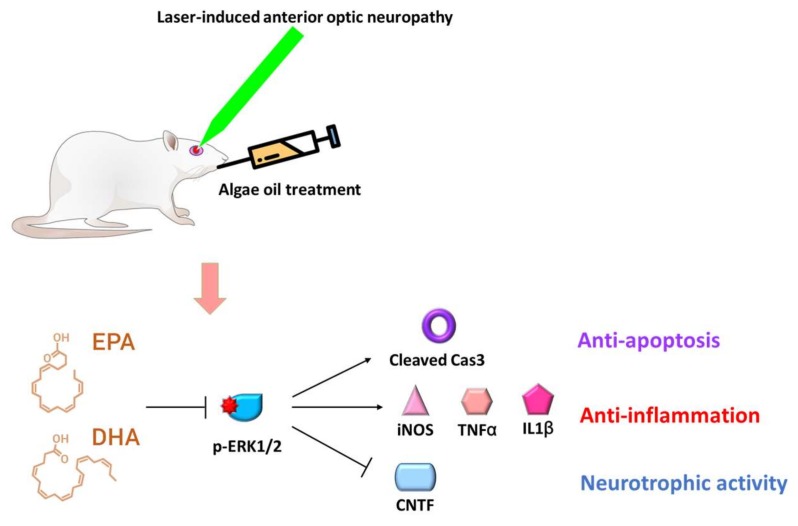
Mechanisms of neuroprotection via multiple pathways in algae oil treatment. DHA-rich algae oil can modulate the expression levels of proinflammatory cytokines, Cl-caspase-3, and antiapoptosis of RGCs, as well as inducing expression of a neurotrophic factor (CTNF) in the retina via the ERK signaling pathway in the rAION model. In addition, ERK inhibition reduced the expression levels of inflammatory cytokines such as iNOS, TNF-α, and IL-1β in the retina.

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
