# Peer review of "Algae Oil Treatment Protects Retinal Ganglion Cells (RGCs) via ERK Signaling Pathway in Experimental Optic Nerve Ischemia"

_marinedrugs, 2020, doi:10.3390/md18020083_

Round 1

Reviewer 1 Report

The manuscript "Algae oil treatment protects retinal ganglion cells (RGCs) via ERK signaling pathway in experimental optic nerve ischemia" analyzes the therapeutic effects of algae oil in protecting RGCs in a rat model of ischemic optic neuropathy (rAION) and deciphers the associated mechanism. The authors adequately justified, performed and described the experiments in the study, which follows on from their work investigating fish oil (The Zone OmegaRx) for 10 days on rAION. A few minor revisions are needed in the paper before publication: 1. A general reader would have trouble following some portions of the paper where abbreviations are not adequately described, like ED1 (marker of CD38 and, therefore, circulating macrophages), AA/EPA (marker of cellular inflammation by following two types of fatty acids), and HPF needs to described in the results in relation to Fig. 4. Please make sure all abbreviations are adequately described in the manuscript, so a general reader can follow the paper more easily. 2. Morphology of the FVEP wavelets need to be described in relation to Fig. 2 - the sense that the shape is generally less maintained in the rAION- than the ALG-group. 3. The TUNEL staining in Fig. 5 would be easier to monitor if the apoptotic cells were, instead of green, falsely-colored red.  4. Minor revisions would be: space in (C)Western in Fig. 4., and formatting 4.5 in Methods for "Retrograde labeling of RGCs with FluoroGold and morphometry of the RGC" with 4.6-4.9 to follow.      

Author Response

Reviewer 1:

The manuscript "Algae oil treatment protects retinal ganglion cells (RGCs) via ERK signaling pathway in experimental optic nerve ischemia" analyzes the therapeutic effects of algae oil in protecting RGCs in a rat model of ischemic optic neuropathy (rAION) and deciphers the associated mechanism. The authors adequately justified, performed and described the experiments in the study, which follows on from their work investigating fish oil (The Zone OmegaRx) for 10 days on rAION. A few minor revisions are needed in the paper before publication:

A general reader would have trouble following some portions of the paper where abbreviations are not adequately described, like ED1 (marker of CD38 and, therefore, circulating macrophages), AA/EPA (marker of cellular inflammation by following two types of fatty acids), and HPF needs to described in the results in relation to Fig. 4. Please make sure all abbreviations are adequately described in the manuscript, so a general reader can follow the paper more easily.

Reply: Thanks for your suggestion to improve the readability of the manuscript. We have modified the sentences in the revised manuscript for describing the abbreviations clearly as following:

In ED1 (marker of CD38, specific for extrinsic macrophages) staining of ON sections, there were 3.1±1.0 ED1+ cells/high power field (HPF) in the sham group. (Page 5, line 119-120).

During the daily gavage with algae oil, the blood AA/EPA (a marker of the cellular inflammation by following two essential fatty acids) ratio gradually decreased from 17.30± 2.1 to 0.76± 0.25 (Figure 1). (Page 3, line 87-88)

Changes in FVEPs (visual function assessment) after rAION were measured at the 4th week after the infarct. (Page 3, line 96-97)

Morphology of the FVEP wavelets need to be described in relation to Fig. 2 - the sense that the shape is generally less maintained in the rAION- than the ALG-group.

Reply: We appreciated that the reviewer provided this useful comment. We have added the sentences in the revised manuscript as following:

The amplitude of the P1-N2 wavelet is used to evaluate the RGC structure and function in vivo. RGC loss results in the P1-N2 amplitude decrease. Previous study showed no significant differences in latency between photoptic and scotopic VEPs in Wistar rats [5]._(Page 3 line 97-99)

The TUNEL staining in Fig. 5 would be easier to monitor if the apoptotic cells were, instead of green, falsely-colored red.

Reply: We agreed with the reviewer’s suggestion. We have changed the color in the Figure 5.

Minor revisions would be: space in (C)Western in Fig. 4., and formatting 4.5 in Methods for "Retrograde labeling of RGCs with FluoroGold and morphometry of the RGC" with 4.6-4.9 to follow.

Reply: Thanks for the reviewer’s correction in this manuscript. We have followed your suggestions to correct them.

(C) Western blot analysis of iNOS, IL-1β, TNF-α and ERK expression of the rat retina after AION at the 3rd day. (page 5, line 139-140)

4.5 Retrograde labeling of RGCs with FluoroGold and morphometry of the RGCs (Page 10 line 292)

4.6 In situ nick end-labeling (TUNEL) assay (Page10 line 304)

4.7 Immunohistochemistry (IHC) of ED1-positive cell in the ON (Page10 line 311)

4.8 Western blotting (Page10 line321)

4.9 Statistical analysis (Page11 line333)

Reviewer 2 Report

The title is suitable.

The text is readable.

The abstract is adequate.

The introduction gives enough information to support the aim of the study, which is clearly stated.

Material and Methods Section is adequate.

Results: experiments are well controlled. The figures are clear.

The list of references is in the correct format.

The discussion is adequate.

In general, this is a thorough and well-done study, and the manuscript is well-written. I suggest that this paper could be suitable for publication. 

Author Response

Reviewer 2:

The title is suitable.

The text is readable.

The abstract is adequate.

The introduction gives enough information to support the aim of the study, which is clearly stated.

Material and Methods Section is adequate.

Results: experiments are well controlled. The figures are clear.

The list of references is in the correct format.

The discussion is adequate.

In general, this is a thorough and well-done study, and the manuscript is well-written. I suggest that this paper could be suitable for publication.

Reply: Thanks for the reviewer 2’s comments and support.